# Optimising Productivity and Safety of the Open Pit Loading and Haulage System with a Surge Loader

Ignacio Andrés Osses Aguayo [1] , Micah Nehring [2,*] and G. M. Wali Ullah [3]

1 Ingeniera Civil de Minas, Universidad de Concepción, Concepción 3349001, Chile; igosses@udec.cl
2 School of Mechanical and Mining Engineering, The University of Queensland, Brisbane, QLD 4072, Australia
3 Department of Mathematics, University of Chittagong, Chattogram 4331, Bangladesh; wali@cu.ac.bd
* Correspondence: m.nehring@uq.edu.au

**Abstract:** The open pit mining load and haul system has been a mainstay of the mining industry for many years. While machines have increased in size and scale and automation has become an important development, there have been few innovations to the actual load and haul process itself in recent times. This research highlights some of the potential productivity and safety benefits that the incorporation of a surge loader may bring to the load and haul system through an analysis of the system, discussion of component characteristics, and mine planning aspects. The incorporation of the surge loader into open pit loading and haulage operations also enables improved safety. This is a result of a reduction in shovel–truck interactions and the reduced likelihood of truck overfilling and uneven loading. This paper details the number of mine worker deaths that a surge loader may have prevented within the Peruvian and Chilean mining industries.

**Keywords:** surge loader; mine safety; load and haul; truck and shovel; open pit mining; haulage systems

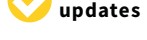



## 1. Introduction

The minerals industry is required to process increasingly lower grade ores [1] in order to meet an insatiable demand for raw materials that is being driven by the general advancement of humanity, and in particular, the global transition toward cleaner forms of energy and its respective storage and transmission. This is occurring against a backdrop of heightened community awareness of environmental, social, and governance issues that have sadly plagued the industry for too long. To meet the challenges associated with having to extract and process greater quantities of ore material at a price that remains reasonable, it is necessary to increase the productivity of operations [2] through innovations.

The safety of operations is essential in the mining industry [3]. This is why the objective of any innovation should also be to increase the safety of operations while lifting their productivity at the same time [3]. Even though it may be challenging, this is the key reason why innovation in the mining industry needs to look beyond small-scale incremental improvements of current systems. Step-change innovation will only occur if new ways to redesign the various stages of the operation are achieved [3].

In current open pit mining operations, one of the highest costs lies in the loading and hauling stage [4]. During the loading and hauling stage, material at the working face is loaded by an excavator/shovel into trucks, which have positioned themselves as best as possible to receive this material. The truck then proceeds to haul this material to either a processing plant, waste dump, or stockpile. Although the current Shovel-Truck (ST) system presents numerous advantages, particularly over inflexible conveyor-based continuous mining systems, it is nevertheless costly and becomes more cost as the operation matures. For this reason, it is vital for the future of open pit operations to find ways to improve the Shovel-Truck system. In this context, the incorporation of a surge loader into this system is worthy of further investigation.



This research highlights some of the potential productivity and safety benefits that the incorporation of a surge loader may bring to the load and haul system through an analysis of the system, discussion of component characteristics, mine planning, and design aspects. While the authors are aware of some in-house studies conducted by various mining companies into the use of surge loaders, very few studies are available in the open literature. This paper thus presents a detailed discussion into the potential productivity and safety benefits that the introduction of a surge loader into the open pit loading and haulage system may present. This is the first of a series of papers that the authors envisage will ultimately delve in-depth into the technical aspects of the use of a surge loader.

## 2. Background

The open pit mining value chain is comprised of initial prospecting and exploration, resource modelling and mine planning, mine production including drilling and blasting followed by loading and hauling, comminution to liberate and separate the valuable mineral, followed by further refining, and finally, transportation to market [5]. In open pit mining operations, the loading and hauling stage is very important in the overall production of the mine. This is because the performance in this stage largely determines the production rate that the mining operation can achieve. Often the loading and hauling stage is the limitation or bottleneck across the whole open pit mining value chain [6]. As such, an efficient, safe, and well-functioning loading and hauling system is essential to maximising mine productivity [7,8] and value for all stakeholders.

The purpose of the loading and hauling stage is to move the material previously fragmented by the drilling and blasting process. The first step in this process consists of loading the material from the bench or working face of the mine into trucks. The next step involves transporting this material to its destination (stockpile, waste dump, or processing plant), via a haul road that generally spirals up the pit walls and is specially designed and maintained to accommodate large haulage trucks of up to a 400-tonne capacity [9].

Depending on operational characteristics and site geometry, the loading and hauling of material typically represent between 35–55% of the operational costs of an open pit mine [10,11]. As open pit mining operations mature and additional resources are discovered, extensions to mine life are common. This results in greater pit depths and thus longer hauls. Ultimately, the cost of material transportation takes a larger and larger share of the operational costs of the site [6,11]. For this reason, one of the challenges for open pit mining operations is to continually optimize the loading and hauling stage. Any improvement could ultimately mean being able to extract additional ore at a greater depth, which would lead to a further increase in the useful life of the open pit mining operation [5].

## 3. Shovel-Truck System

Currently, the most common system for the loading and hauling stage in open pit mining is the Shovel-Truck system (ST) [9,12]. This comprises of a shovel loading blasted material into a truck, which transports the material from the dig face to a destination (stockpile, dump, or primary crusher) where it is unloaded. The truck then returns to the shovel and the cycle repeats, as shown in Figure 1. This has been the mainstay of open pit mining for many years. This system is simple and easy to implement in mining operations and is considered as being very reliable, flexible, and effective [9,13]. One of the main advantages of this system is its versatility as it only needs roads suitable for the movement of trucks without the need for more complex infrastructure such as conveyors. A substitute truck may also be rapidly dispatched to replace any breakdown. This allows the system to easily change and fit to the design of the mine and is thus more likely to prevent the Shovel-Truck system from becoming a limitation in any mine production expansion [14,15].

The Shovel-Truck system has undergone multiple improvements over time. One of these improvements involves optimization of the travel route, whereby the most efficient route for the system is selected through production and topographic analysis. This reduces cycle time, transport cost, and increases the productivity of the system [16,17]. All these

changes have improved the productivity of the system as a whole, however, they have not altered the cycle itself (Figure 1) and thus only generate incremental improvements.

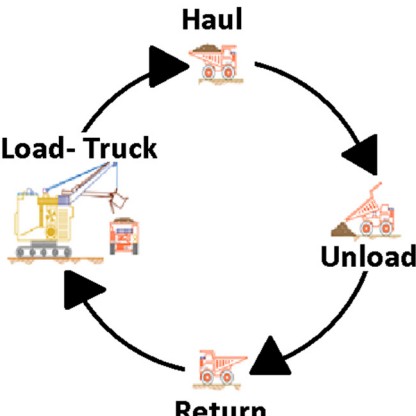

**Figure 1.** Cycle of the Shovel-Truck system.

Although the Shovel-Truck system has proven its functionality over the years, some safety aspects remain [18–20]. The various interactions between the shovel and the truck also lack precise control over the truck's fill factor.

### 4. Surge Loader System

While the concept of the surge loader has been in existence for many years, it is only recently that an equipment manufacturer is offering this as a standard product to the broader traditional open pit mining industry. In conjunction with the further development of scanning and sensing technologies, the latest surge loader now has additional capabilities that it previously did not. This now makes the surge loader a new and exciting proposition for many open pit mining operations around the world. As with any new potential equipment purchase, a thorough study should be conducted to understand if the ongoing additional revenue as a result of productivity improvements is able to offset the initial substantial capital cost associated with the purchase and commissioning of a surge loader.

In order to introduce this piece of equipment and due to a lack of alternatives, this paper contains illustrations mostly of the 'Fully Mobile Surge Loader' from MMD. The MMD Fully Mobile Surge Loader is the first of its kind in the world. It is designed to revolutionize the loading of haul trucks; making the process faster, more efficient, and safer. It should be noted that the authors have no affiliation with this product or its manufacturer.

The surge loader is a device designed for use in the loading process, whose main function is to receive the material from the shovel and to then load trucks [21]. It thus serves as an intermediary between the shovel and the truck. The result is an increase in the safety and productivity of the loading and hauling stage, as it divides the cycle of the Shovel-Truck system into two independent cycles [21] as shown in Figure 2.

The cycle of the Shovel-Truck system is divided into two cycles as a result of the surge loader. This means that the shovel and the truck no longer directly interact. Rather, the surge loader acts as an intermediary, which allows functional independence between the loading machine and trucks, respectively.

The surge loader consists of a hopper with a capacity that is generally about 2.5 times (but not always) the capacity of the truck used in the mining operation [22]. The hopper receives material from the shovel which it then transfers to a truck, as shown in Figure 3. With the surge loader, the shovel no longer depends on the immediate presence of a truck to complete its cycle. The continued operation of the shovel now only depends on the remaining available capacity of the hopper. This eliminates direct dependence between the shovel and the truck that exists in the classic Shovel-Truck system [23]. The hopper is track mounted which gives it the mobility and ability to be completely autonomous [23] and to accompany the shovel under normal bench operating conditions [21]. This is a

great advantage when compared to other loading and transport systems used in open pit mining with limited mobility and autonomy, such as In-pit Crusher and Conveyor (IPCC) systems [24]. However, it must also be recognized that the surge loader is an additional item of equipment that may breakdown and require maintenance. While planned maintenance may take place to minimize the disruption to productivity as much as possible, any unplanned maintenance as a result of a breakdown is likely to cause a larger disruption to production than it otherwise would.

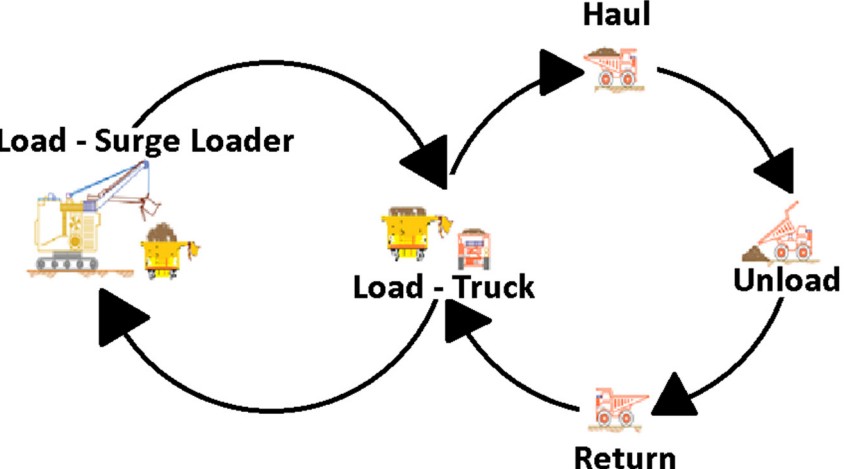

**Figure 2.** New Shovel-Truck system.

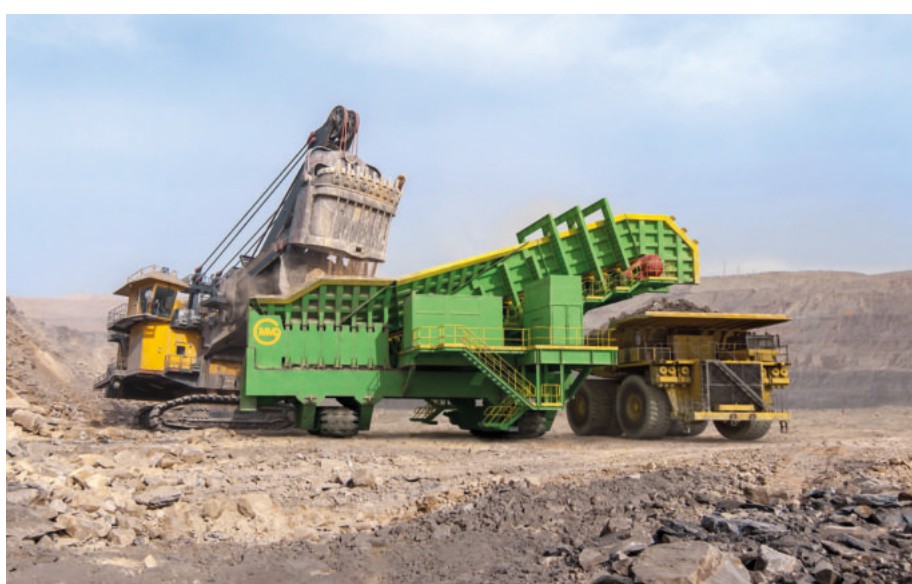

**Figure 3.** Fully Mobile Surge Loader [21].

### 4.1. Feeding System

The surge loader feeding system utilises a 'Heavy Duty Apron Plate Feeder' [23]. Figure 4 shows a surge loader feeding system in operation within a coal mine in Colombia. This installation has made it possible for the manufacturer to test its functionality under different operational scenarios, in addition to proving its efficiency in the loading process [23].

The feeding system consists of a conveyor belt that carries the material from the hopper to the truck. The use of the conveyor belt allows the average filling of 330-tonne trucks in 60 s [23]. This represents a significant reduction of approximately 50% in the time taken to load trucks compared to the traditional Shovel-Truck system [1].

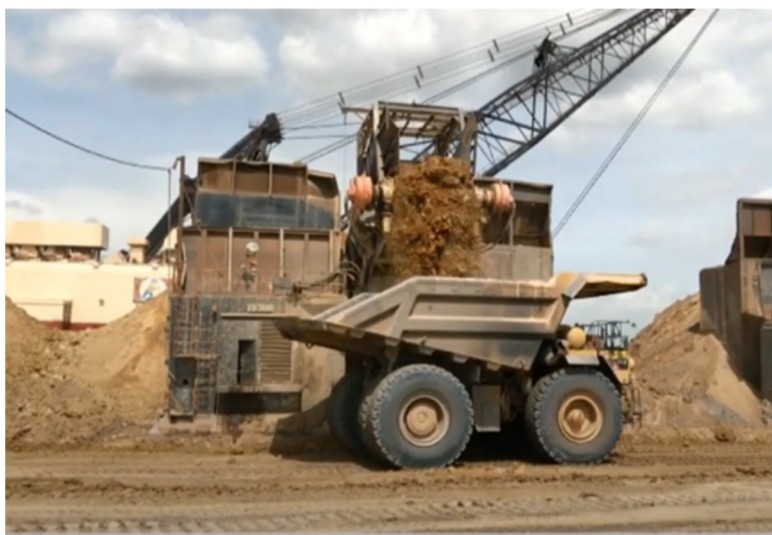

**Figure 4.** Feeding and Loading equipment (Colombia) [23].

The feeding system is comprised of high resistance plates, designed to handle high impact and abrasive materials [25]. These plates are designed for low maintenance requirements, long useful life, and robustness and reliability. They feature overlapping edges which prevent spillage between the plates and are fixed to chains with bolts that are positioned between the grousers, protecting the bolt heads from damage caused by the material being conveyed as illustrated in Figure 5 [25].

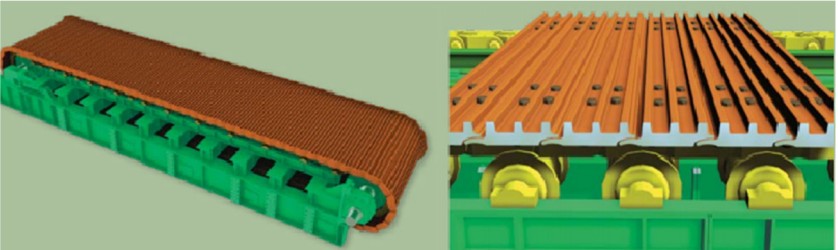

**Figure 5.** Conveyor belt [25].

The main features of the apron plate feeder are the heavy-duty chains and rollers (manufactured by Caterpillar as part of the MMD system), which are attached to the main frame, as shown in Figure 6. These stand out for their high resistance and elimination of impact energy, which is initially absorbed by the conveyor plates, by deforming within their elastic limits. The impact rails then transmit the forces which are dissipated into the main frame construction.

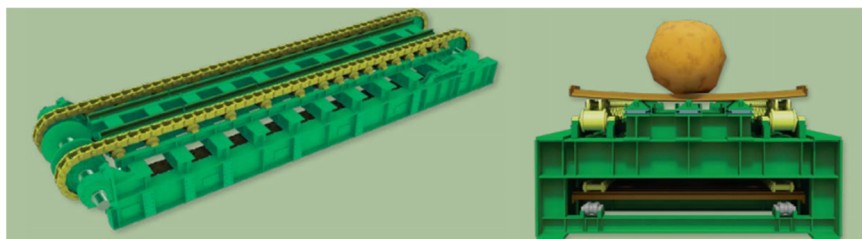

**Figure 6.** Mobility and impact methods of the feeding system [25].

*4.2. Receiving and Loading of the Material*

The transfer of material between the surge loader and the truck is carried out through a conveyor belt that carries the material previously deposited into the hopper by the shovel

to the truck as shown in Figure 7. The belt, through strategically placed sensors, is capable of measuring the volume and weight of the material that is being delivered, managing to accurately control the fill factor of the truck [20,21]. Through these sensors, the surge loader is capable of detecting the presence of large rocks which can affect how this material is transported [23,26,27]. Together with the information of the material already loaded into the truck, this allows the monitoring system to decide if the truck is capable of transporting this larger rock without exceeding safety limitations [23]. In this case, if the result is negative, the surge loader stops loading and gives the signal for the truck to continue its cycle with the material that was already loaded. In addition to this, the system leaves enough material between the large rock and the material discharge point to serve as a cushion for the impact of the large rock discharging (Figure 8) into the truck tray. The on-board sensor system gives the surge loader a great advantage over the conventional material transfer system (Shovel-Truck system), by accurately controlling the fill factor of the truck and significantly narrowing the fill factor variance [22]. Controlling the fill factor in the classic system (Shovel-Truck system) is carried out by appropriately matching the capacities of the shovel and the truck. Fill factors in the classic system are also influenced by manoeuvrability, the distribution of the material in the shovel bucket, material fragmentation, and operator competence [28]. The effects of many of these items are diminished with the incorporation of a surge loader.

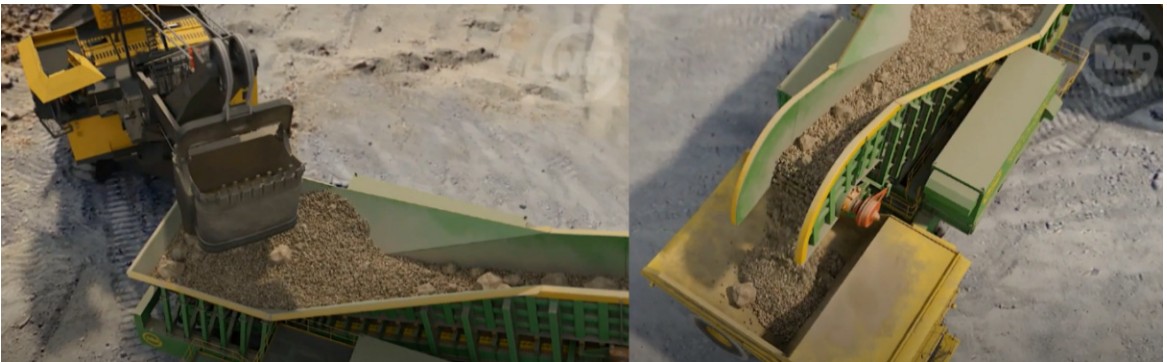

**Figure 7.** Material transfer sequence [23].

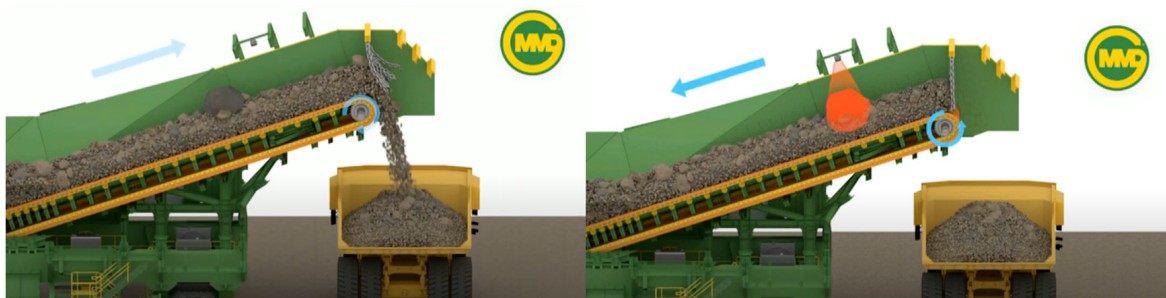

**Figure 8.** Presence of a large rock on the conveyor belt [23].

From a review of its components, it is evident that a surge loader is a large piece of equipment and requires a significant geometrical footprint. As a result of this, the mine design and subsequent plan may have to be altered in order to accommodate this machine.

## 5. Effect of the Surge Loader on Truck Productivity

In the classic Shovel-Truck system (without the inclusion of the surge loader) the truck fill factor depends entirely on the shovel and how much material it can load into the truck. This generates three possible scenarios. The first is when the truck exceeds its

payload (overfill), which generates an increase in the risk associated with the transport of material [18–20]. For this reason, the on-board measurement system seeks to avoid this and promptly cuts the material supply from the transfer conveyor. The second scenario is when the opposite occurs and the truck is loaded with less than its payload (underfill). This scenario is the most common and causes a decrease in the designed productivity of the truck. The third scenario is where the truck is loaded with its exact payload capacity. This scenario is optimal, but due to different factors such as the loading capacity of the shovel or the competence of the operator, it is very difficult to obtain. This makes the choice of shovels and trucks dependent on each other from a productivity viewpoint. Shovels that can load trucks in as few cycles as possible is thus favoured as it minimises the deviation in the truck fill factor.

The surge loader, through its feeding system, allows for the controlled loading of material into the truck. This is not only practical from the point of view of reducing the loading time, but the feeding system also allows for the control of the fill factor, which is now not dependant on the loading capacity of the shovel but rather on the payload of the truck. This makes it possible to consistently achieve filling factors as close to 100% as possible [23]. This, in turn, allows operations to close the gap on the ideal scenario, which is to achieve maximum truck productivity safely and without increasing the cost of haulage. Given that the truck fill factor of the classic Shovel-Truck system tends to be approximately 90% on average [29], raising this to close to 100% represents a significant potential improvement in productivity for the same number of trucks. In some cases, this productivity improvement may even require a smaller trucking fleet. For other operations, improved trucking productivity may change the production bottleneck from the mine to another aspect of the operation.

The use of the surge loader also results in shovels and trucks operating independently of each other, since the surge loader eliminates the interaction between this equipment. This independence potentially allows for a greater range of shovels and trucks that could be utilized in the operation. This allows the option of having different types and capacities of trucks operating together to undertake the hauling of material, which allows for a more dynamic loading and hauling stage that may better adapt to the different types of materials in the mine at the various stages of extraction.

## 6. Safety Features

In recent years, great efforts have been made to improve the health and safety of mine workers through innovation in the methods and machines used in mining operations [3,30,31].

### 6.1. Sensors and Cameras System

The use of autonomous equipment is increasingly common in mining. The surge loader [3] can also operate autonomously. For autonomous vehicles to operate successfully they need to be aware of the distance between themselves and other vehicles with enough time to make safe and reliable mission plans [32,33]. The use of cameras and sensors are therefore not only essential for the correct operation of these vehicles and equipment but to also maintain the high standards of safety necessary in the mining industry [18,20].

The surge loader uses a network of sensors and cameras to detect their surroundings including the approach and departure of trucks. This system determines when a truck is approaching the loading point as shown in Figure 9. Then, using Radio Frequency Identification (RFID) (RFID uses electromagnetic fields to automatically identify and track tags attached to objects) sensors, it is capable of detecting when the truck is at the exact loading point required for the surge loader. As shown in Figure 10, a signal is sent to the truck to stop. Finally, the surge loader returns a signal to the truck when it is loaded to continue its journey. This detection system is completely computerized and both cameras and sensors are remotely controlled. This allows for the material transport process to be automated and eliminates the need for a human operator inside the truck to determine the stopping point.

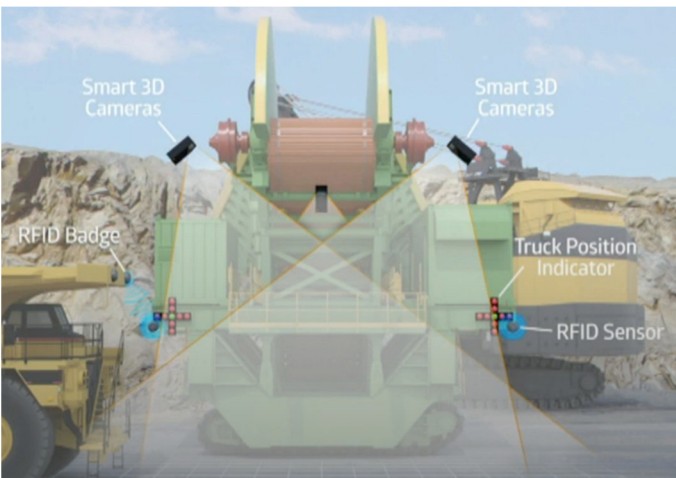

**Figure 9.** Camera system [23].

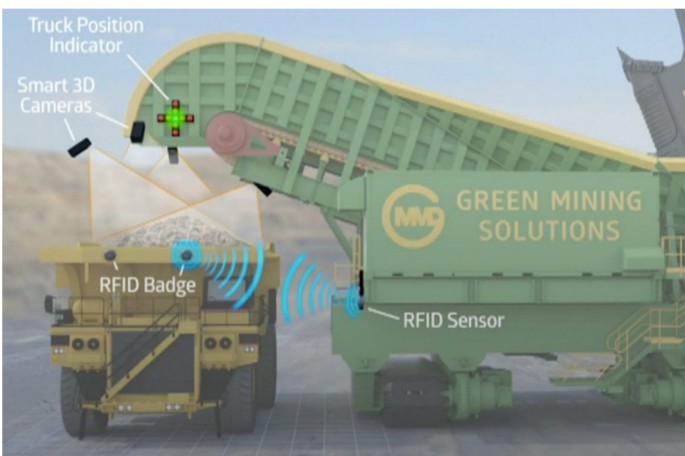

**Figure 10.** Sensor system [23].

While the use of sensor and scanning technology is no doubt having profound impacts across many industries, the environment in which these are used in mining operations is often prone to dust, which may hinder the full capability of some sensors and scanners.

*6.2. Travel Routes*

The routes that trucks take in the material transport process of the Shovel-Truck system are an essential aspect since they influence the productivity achieved by the truck. That is why it is important to work with optimal routes that improve the productivity of the truck [17]. In the classic Shovel-Truck system, however, although truck routes are optimized, there are 'dead' sections where the transportation process is delayed [34] and these sections cannot be eliminated. One of these sections corresponds to the manoeuvres carried out by the truck to locate itself appropriately at the loading point [34].

The surge loader eliminates the need for the truck to maneuver multiple times to be located appropriately near the shovel. This is a very common process in the classic Shovel-Truck system. The use of cameras and sensors mounted on the surge loader facilitates the automation of trucks by enabling an intelligent communication system between the surge loader and the truck. In addition, trucks will take simpler routes (without positioning maneuvers) often by facilitating the drive-by-loading method as shown on the right-hand-side of Figure 11. The inclusion of the surge loader to the Shovel-Truck system thus allows for working with more continuous loading routes. On these new routes, manoeuvring times are largely eliminated. In addition to reducing truck loading time, this reduces the overall truck cycle time.

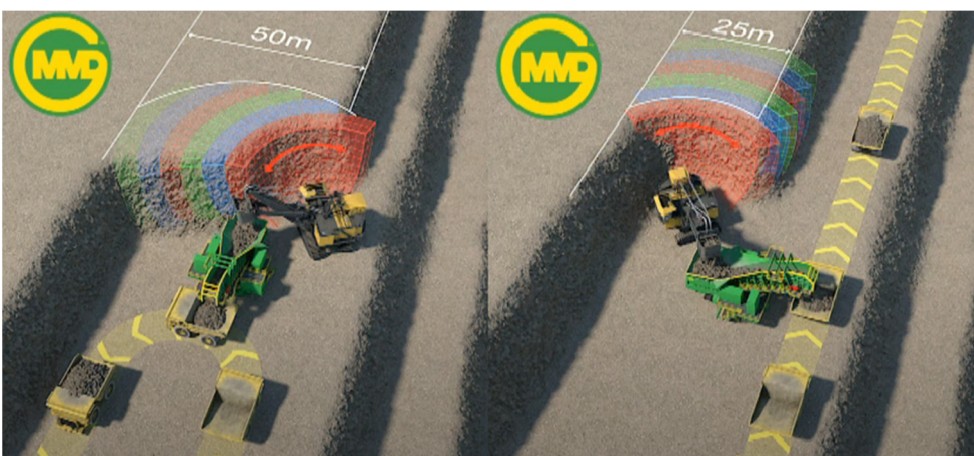

**Figure 11.** Truck routes with the surge loader [23].

The proximity detection system of the surge loader reduces operational maneuvers of trucks (turns, reversing, stopping, or exiting the route), due to the ability to detect and eliminate blind spots [32,35]. The decrease in the probability of collisions during normal driving, parking, or maintenance maneuvers is another benefit associated with the detection system. Consequently, this results in extensions of the useful life of rims, tires, and suspension systems [32,35]. The main benefit of the detection system is an increase in safety (by preventing accidents). Other benefits likely include reduced maintenance costs [32,35]. While these benefits are not direct improvements to the loading and hauling stage, they are benefits that affect the mine, its financial viability, and the planning process.

*6.3. Safety*

Increasing safety is a priority in modern mining operations [19,29,31]. One of the current challenges is to find new ways to achieve greater levels of safety without, or, with minimal reduction, in the productivity of operations [5]. In this context, one of the main risks present in open pit mining is the loading and hauling stage [18–20,36,37], since the Shovel-Truck system (typically used in this stage) requires direct interaction of large equipment [36]. In general, any accident that occurs in the loading and haulage stage results in a delay and decrease in mine productivity [36]. Considering that in open pit mining the loading and haulage stage determines the productivity of the operation [5], it is very important in this type of operation to keep accident rates to a minimum [31,36].

On the world stage in the field of mining, Peru and Chile are both leaders in the production of vital metals including copper, gold, manganese, and zinc. Both have positioned themselves as leaders, which has been achieved through their mining policies, high export rates, and their large number of active mines. Records of fatal accidents occurring in the mining industries of Peru and Chile are tabulated in Tables 1 and 2, respectively.

Table 1 shows the fatal mining accidents registered by the Ministry of Energy and Mines (MINEM) of Peru between 2000 and 2016. It can be seen that 41 accidents out of 842 correspond to accidents that occurred in the loading and haulage stage.

While the number of accidents may seem high it should be noted that most of the registered accidents are concentrated in the first seven years of the sample (2000–2007). As shown in Table 1, the largest source of recorded accidents corresponds to accidents related to falling rocks (representing 32%). Although the loading and hauling stage is not the main source of accidents, this activity represents 5% of the total registered, which is still very important considering that these are accidents involving deaths.

In the case of Chile, the records of the Chilean National Geology and Mining Service (SERNAGEOMIN), observed in Table 2, show that between 2010 and 2019, Chile recorded 228 accidents in the mining industry, of which 41 correspond to accidents that occurred in open pit mining.

**Table 1.** Number of accidents resulting in death in Peru (2000–2016).

| Type of Accident | Number of Accidents |
|---|---|
| Rock fall | 271 |
| Fall of workers | 82 |
| Vehicle traffic | 76 |
| Others | 65 |
| Landslide | 63 |
| Intoxication and suffocation | 70 |
| Loading and haulage | 41 |
| Explosions | 33 |
| Equipment maneuvering | 54 |
| Electric energy | 38 |
| Material handling | 19 |
| Burial by subsidence of land | 23 |
| Tools | 7 |
| Total | 842 |

Source: Ministry of Energy and Mines (MINEM), Government of Peru.

**Table 2.** Number of accidents resulting in death in Chile (2010–2019).

| Location | Number of Accidents |
|---|---|
| Underground mine | 117 |
| Open pit mine | 41 |
| Port | 2 |
| Workshops | 4 |
| Others | 7 |
| Road | 17 |
| Tailings dump | 4 |
| Processing plant | 30 |
| Surface installation | 6 |
| Total | 228 |

Source: Geology and Mining Service (SERNAGEOMIN), Government of Chile.

Although the highest number of registered accidents corresponds to accidents related to underground mining (more than double the accidents than open pit mining operations), the second-highest source of accidents corresponds to accidents that occurred in open pit mining, representing 18% of the total. As in the case of Peru, the recorded accidents are fatal so this is still very significant.

The various operational mechanisms that the surge loader would negate, including (1) loading the truck without it approaching the shovel, (2) not needing to perform additional maneuvers to accommodate itself, (3) controlling the loading of the material to avoid overfilling, and (4) the transportation of poorly balanced loads due to the presence of large rocks, would increase the safety of the loading and haulage stage of open pit mining operations. Table 3 shows the data in Tables 1 and 2 broken down into further categories. It can be observed that, of the fatal accidents mentioned previously, 27 accidents in the case of Peru could have been avoided. This represents 66% of all accidents related to the loading and haulage stage, with fourteen accidents avoidable with the use of the feeding system, six accidents with the use of the new routes, and seven with the surge loader as an intermediary step (Table 3). In the case of Chile, the use of the surge loader would have

prevented seven fatal accidents (Table 3). This number can be perceived as low, but, it represents 17% of the open pit mine accidents, with four accidents being avoidable with the use of the feeding system, one with the use of new routes, and two with the surge loader as an intermediary step. The use of the surge loader could have avoided around 3% of the total fatal accidents in each registry (Peru: 3.21%, Chile: 3.07%). This percentage represents accidents related to truck rollovers on the road because of overfill accidents that occurred due to inappropriate positioning of the truck at the loading point.

**Table 3.** Avoidable accidents.

| Open Pit Mine Accidents | Number Accidents PERU | Number Accidents Chile | |
|---|---|---|---|
| Truck accidents due to overfill | 14 | 4 | Avoidable with the use of the surge loader |
| Truck and Shovel crash accidents | 6 | 1 | |
| Accidents in the process of loading material from the shovel to the truck | 7 | 2 | |
| Not avoidable with the use of the surge loader | 14 | 34 | |
| Total | 41 | 41 | |

Source: Ministry of Energy and Mines (MINEM), Government of Peru, Geology and Mining Service (SERNAGEOMIN), Government of Chile.

The main aspect of the surge loader that would avoid most of these accidents is its feeding system. This prevents the overfilling of trucks, which, as can be seen from the data recorded in Peru and Chile, is one of the leading reasons for fatal accidents in the loading and haulage stage.

## 7. Conclusions

In conclusion, adding equipment such as the surge loader to the Shovel-Truck system allows for significant improvement in the production cycle of the loading and hauling stage in open pit mines. The surge loader not only allows for productive independence of both the shovel and the truck but also allows for greater freedom of choice in the selection of trucks. With the incorporation of the surge loader, the shovel and the truck no longer have to maintain a certain match in their capacities. This broadens the catalog of equipment available for the loading and hauling stage.

The surge loader makes it possible to simplify the travel routes of the trucks, which speeds up the transport of material by reducing maneuvering times. In addition, the material unloading system of the surge loader allows for control over the fill factor of the trucks, which increases the efficiency of the loading process. For these reasons, the new cycle that the surge loader applies not only separates the shovel cycle from the truck cycle but also improves the loading and hauling stage.

From a safety point of view, research shows that the different features of the surge loader decrease the risks of the loading and hauling stage by avoiding interactions between the equipment. Its feeding system also manages better control over the fill factor of the trucks than present Shovel-Truck systems. This avoids any possibility of generating an overfill of the truck which increases the risk of accidents as seen in the fatal accident data of Chile and Peru. Its incorporation could have prevented 3% of the fatal mining accidents that occurred in Chile and Peru.

Although there is still very limited data on the operation of this equipment in the field, both its characteristics and the analysis of the accident data from Chile and Peru show that adding the surge loader to the Shovel-Truck system is an innovation that would improve both the productivity and the safety of the loading and hauling stage. The modern surge loader, therefore, represents a potentially new way for future loading and haulage that would be fundamental for the future of open pit mining.

## 8. Future Research

Several of the aspects of the surge loader requires further investigation. First and foremost, a detailed simulation for the new loading and haulage system that incorporates the surge loader should be undertaken to determine the true impact on productivity. This might consist of a simulation of the shovel productivity and the truck fleet productivity in isolation from each other followed by a simulation that combines the entire loading and haulage fleet.

While conventional thinking suggests that the capacity of the surge loader should be 2.5 times the capacity of the truck, it is yet to be proven in the literature that this is optimal. Another simulation should therefore alter the size of the surge capacity to identify its impact. In addition to this, a mixed truck fleet of varying capacities should also be investigated to determine this impact.

The size and footprint of a surge loader also need to be taken into account in the mine planning and design process. Likely, the deployment of a surge loader onto an operating bench within an open cut mine could warrant the use of a wider pushback. If this is the case, a redesign and of the mine will be required, which will have follow-on impacts on productivity and financial metrics.

While scanning and sensor technology has improved immensely, the impact of dust on the potential disruption of scanners and sensors to prevent adequate communication between equipment should be fully investigated. The dust that can be generated in an open pit mining environment can be significant. It is therefore important that the most suitable type of scanning and sensing technology is used.

While the surge loader reduces several of the typical risks associated with conventional Shovel-Truck loading and haulage, a detailed and thorough investigation should take place to identify any additional risks that the surge loader may introduce into the system. A detailed risk assessment should also determine the consequences and likelihood of these and identify potential mitigation strategies.

**Author Contributions:** All authors have contributed equally to the conceptualization, literature review, analysis and drafting of this manuscript. All authors have read and agreed to the published version of the manuscript.

**Funding:** This research received no external funding.

**Conflicts of Interest:** The authors declare no conflict of interest.

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
