# Peer review of "Optimising Productivity and Safety of the Open Pit Loading and Haulage System with a Surge Loader"

_mining, doi:10.3390/mining1020011_

Round 1

Reviewer 1 Report

The manuscript focuses its attention on presenting recent equipment, the “surge loader”, produced by a company in the market. The presented equipment allows the division of the loading and hauling stage into two stages with reduced interdependence.

Two-stage separation has advantages from a conceptual point of view. However, throughout the manuscript, in addition to the explicit references to an equipment manufacturer, only the advantages are presented. The authors do not present the disadvantages of the system nor the conditions of its implementation. It is not shown any related costs or any other factual data that would allow anyone to base their choice to implement this new approach.

Under these conditions, but taking into account the practical interest of the subject, it is suggested the authors improving and resubmit the article. For that, They must, at least, remove references to the equipment manufacturer and detail the raised issues.

Their opinions must be based on objective elements so that their proposal has an effective interest in the technical and scientific community.

Most references originate from grey literature or are very incompletely referenced. Thus, the references also need to be improved.

Author Response

The manuscript focuses its attention on presenting recent equipment, the “surge loader”, produced by a company in the market. The presented equipment allows the division of the loading and hauling stage into two stages with reduced interdependence.

Response: Thank you. No response required.

Two-stage separation has advantages from a conceptual point of view. However, throughout the manuscript, in addition to the explicit references to an equipment manufacturer, only the advantages are presented. The authors do not present the disadvantages of the system nor the conditions of its implementation. It is not shown any related costs or any other factual data that would allow anyone to base their choice to implement this new approach.

Response: The authors have expanded section 8. Future Research to discuss some of the aspects requiring further attention in any potential implementation of a surge loader, including the impacts of a wider pushback on the mine design and the potential of dust to disrupt the scanning and sensing technology. The scope of this paper is not to devolve into the cost aspects of the surge loader but rather to focus on potential improvements to productivity and safety. Each mining operation is different – so an independent detailed study will need to be carried out on each anyway.

Under these conditions, but taking into account the practical interest of the subject, it is suggested the authors improving and resubmit the article. For that, They must, at least, remove references to the equipment manufacturer and detail the raised issues.

Response: unfortunately there is very limited additional information available on the surge loader. The authors therefore had to reference the only known supplier of this technology as a main stream product. The authors have been very open and transparent about this and had already included the following sentences at the beginning of section 4:

‘While the concept of the surge loader has been in exitance for many years, it is only recently that an equipment manufacturer is offering this as a standard product to the broader traditional open pit mining industry. In order to induce this piece of equipment and due to a lack of alternatives, this paper contains illustrations mostly of the ‘Fully Mobile Surge Loader’ from MMD.’

Their opinions must be based on objective elements so that their proposal has an effective interest in the technical and scientific community.

Response: The authors agree. This is the first of a series of papers that are planned on this topic. As such, the purpose of this paper is to introduce the topic, the equipment and provide a general discussion around potential benefits. Further papers will effectively address the items listed in section 8. Future Research, which in future includes a full simulation of the system.

Most references originate from grey literature or are very incompletely referenced. Thus, the references also need to be improved.

Response: The editorial office will help fix this.

Reviewer 2 Report

The references format must be revised according to https://www.mdpi.com/authors/references

There are some error messages that must be solved (see row 88, 104, 122, 133, …)

It is stated that introducing the surge loader have a number of advantages:

  • reduction of about 50% in the time taken to load trucks
  • the feeding system ensure an increase fill factor from 90 % close to 100%
  • decoupling the shovel from the trucks allows a greater range of shovels and trucks, improve the safety of mine, eliminates the need for the truck to maneuver multiple times
  • enable the use of autonomous equipment due to a network of sensor and cameras

I consider it necessary to supplement the article with the influence of dust on the network of sensors and cameras and the implications on productivity and safety, that will balance the list of only benefits presented by the equipment manufacturer.

Author Response

The references format must be revised according to https://www.mdpi.com/authors/references

Response: The editorial office will help fix this.

There are some error messages that must be solved (see row 88, 104, 122, 133, …)

Response: Thank you. These, and others have been addressed.

It is stated that introducing the surge loader have a number of advantages:

  • reduction of about 50% in the time taken to load trucks
  • the feeding system ensure an increase fill factor from 90 % close to 100%
  • decoupling the shovel from the trucks allows a greater range of shovels and trucks, improve the safety of mine, eliminates the need for the truck to maneuver multiple times
  • enable the use of autonomous equipment due to a network of sensor and cameras

I consider it necessary to supplement the article with the influence of dust on the network of sensors and cameras and the implications on productivity and safety, that will balance the list of only benefits presented by the equipment manufacturer.

Response: The authors have expanded section 8. Future Research to discuss some of the aspects requiring further attention in any potential implementation of a surge loader, including the impacts of a wider pushback on the mine design and the potential of dust to disrupt the scanning and sensing technology.

Reviewer 3 Report

  1. Please check the paper. Instead of figure number, the message appears (Error! Reference source not found).
  2. Figures 1 and 2 are elementary; extremely simplistic for the journal level.
  3. The idea of a surge loader is not a new one; in principle it is not clear what the original contribution of the authors is.
  4. I ask the authors to specify what is their contribution regarding the introducing of surge loader in the excavation-load process, including the monitoring with cameras and sensors.
  5. To what extent is the productivity of the excavation-load process affected in the conditions of the appearance of an additional process?

Author Response

1. Please check the paper. Instead of figure number, the message appears (Error! Reference source not found).

Response: The editorial office will help fix this.

2. Figures 1 and 2 are elementary; extremely simplistic for the journal level.

Response: These figures maybe elementary, but they are absolutely essential to explaining the fundamental topic of this paper.

3. The idea of a surge loader is not a new one; in principle it is not clear what the original contribution of the authors is.

Response: That’s right, the idea of the surge loader itself is not new, but a surge loader coupled with the latest scanning and sensing technology is. The additional benefits that the latest scanning and sensing technology brings that a surge loader on its own cannot as highlighted in the paper include:

  • rapid truck loading time.
  • better fill factors - close to 100%
  • eliminates the need for the truck to manoeuvre multiple times
  • better reduction in uneven loading
  • better handling of oversize materials

4. I ask the authors to specify what is their contribution regarding the introducing of surge loader in the excavation-load process, including the monitoring with cameras and sensors.

Response: This is the first of a series of papers that are planned on this topic. As such, the purpose of this paper is to introduce the topic, the equipment and provide a general discussion around potential benefits. Further papers will effectively address the items listed in section 8. Future Research, which in future includes a full simulation of the system, and thus provide an accurate reflection of the increase in productivity for a given system. The main contribution of this paper are the statistics that are presented which highlight the potential reduction in deaths had a surge loader been incorporated into the system.

5. To what extent is the productivity of the excavation-load process affected in the conditions of the appearance of an additional process?

Response: As the paper suggests, the inclusion of the surge loader provides an almost complete de-coupling between the loader process and the hauling process. The excavation process is no longer restricted by the loading/hauling process.

Round 2

Reviewer 1 Report

Like any other equipment and/or technology, the surge loader has advantages and disadvantages, some of which are mentioned in "Future Research". These problems should be placed in opposition to the mentioned advantages in a previous chapter. This is regardless of whether or not they can be considered for future research.

As for the bibliography, it must be effectively improved. It's not a matter of formatting.

Of the 27 references, only 11 can be considered peer-reviewed (not grey literature), and of these, 2 are conference articles.

Of the 9 publications in peer-reviewed journals (33%), only 3 (11%) are less than 5 years old.

This scarcity of up-to-date bibliography on the subject of loading and haulage systems raises some doubts as to whether this new system is truly being compared with the state of the art of traditional loading and haulage systems. This question is raised regardless of the interest and advantages that the new system may have. In other words, authors should take a more scientific approach when they intend to publish in a scientific journal.

Author Response

Like any other equipment and/or technology, the surge loader has advantages and disadvantages, some of which are mentioned in "Future Research". These problems should be placed in opposition to the mentioned advantages in a previous chapter. This is regardless of whether or not they can be considered for future research.

Response: The authors have restructured the paper to also include the disadvantages of the surge loader while presenting the advantages. The following paragraphs have been added:

Lines 115-119: As with any new potential equipment purchase, a thorough study should be conducted to understand if the ongoing additional revenue as a result of productivity improvements is able to offset the initial substantial capital cost associated with the purchase and commissioning of a surge loader.

Lines 148-152: However, it must also be recognized that the surge loader is an additional item of equipment that may breakdown and require its own maintenance. While planned maintenance may take place to minimize disruption to productivity as much as possible, any unplanned maintenance as a result of a breakdown will cause a larger disruption to production then it otherwise would.

Lines 210-212: From a review of its components, it is evident that a surge loader is a large piece of equipment and requires a significant geometrical footprint. As a result of this, the mine design and subsequent plan may have to be altered in order to accommodate this machine.

Lines 275-277: While the use of sensor and scanning technology is no doubt having profound impacts across many indurties, the environment in which these are used in mining operations is often prone to dust, which may hinder their fully capability.

As for the bibliography, it must be effectively improved. It's not a matter of formatting.

Of the 27 references, only 11 can be considered peer-reviewed (not grey literature), and of these, 2 are conference articles.

Of the 9 publications in peer-reviewed journals (33%), only 3 (11%) are less than 5 years old.

This scarcity of up-to-date bibliography on the subject of loading and haulage systems raises some doubts as to whether this new system is truly being compared with the state of the art of traditional loading and haulage systems. This question is raised regardless of the interest and advantages that the new system may have. In other words, authors should take a more scientific approach when they intend to publish in a scientific journal.

Response: It is extremely difficult to find research works directly relating to the surge loader in open literature (especially high-quality journal articles). This is because this is such an under developed field of research. This topic therefore warrants further research and papers such as this. If the reviewers can suggest additional references relating directly to surge loading systems the authors would be really appreciative and happy to incorporate these into the paper.

In any case, the authors have added an additional 10 references relating to general truck and shovel operations to further back up existing lines of reasoning within the paper. Most of the new references are from 2017 on-wards from published articles or books. The paper now has a total of 37 references.

Reviewer 2 Report

In this form, the article have an information caracter rather than a scientific one, ​due to the lack of measured data regarding loading an hauling time cycle, availability of equipment, productivity, etc.

Author Response

In this form, the article has an information character rather than a scientific one, ​due to the lack of measured data regarding loading and hauling time cycle, availability of equipment, productivity, etc.

Response: Yes, this is correct. The aim of the paper is to review the newly developed surge loader together with all of its in-built scanning and sensing technology which has the potential to increase productivity and safety. Some safety statistics are presented which highlight the incidents in Chile and Peru that could potentially have been avoided using the surge loader.

Another follow-on paper currently under development delves into the productivity aspects of incorporating a surge loader. This follow-on paper will present the results of a simulation while taking cycle times, availabilities into consideration. However, the authors still believe that there is sufficient technical and scientific detail in this paper to warrant publication in its own right.

Round 3

Reviewer 1 Report

No more comments.